# Preoperative Short-Course Radiotherapy and Surgery versus Surgery Alone for Patients with Rectal Cancer: A Propensity Score-Matched Analysis at 18-Year Follow-Up

**DOI:** 10.3390/biomedicines9070725

**Published:** 2021-06-24

**Authors:** Radoslaw Pach, Piotr Richter, Marek Sierzega, Natalia Papp, Antoni Szczepanik

**Affiliations:** 1Department of General, Oncological, Gastrointestinal and Transplantation Surgery, Medical College, Jagiellonian University, 30-688 Krakow, Poland; piotr.richter@uj.edu.pl (P.R.); marek.sierzega@uj.edu.pl (M.S.); antoni.szczepanik@uj.edu.pl (A.S.); 2Faculty of Medicine, Poznan University of Medical Sciences, 60-512 Poznań, Poland; pappnata00@gmail.com

**Keywords:** rectal cancer, preoperative radiotherapy, overall survival, recurrence free survival, second primary neoplasms

## Abstract

A significant problem for long-term rectal cancer survivors may be the late toxicity of radiotherapy. It creates the possible risk of developing second primary malignancy and a theoretical decrease in overall survival. This study aimed to assess the influence of short-course preoperative radiotherapy in patients with locally advanced rectal cancer on overall survival, local recurrence rate, and second malignancy at 18-year follow-up. The rectal cancer trial was conducted in a single tertiary center between February 1992 and June 2006. A total of 389 patients with locally advanced rectal cancer (cT2-cT4, cN0/+, cM0) were included in the study. Preoperative radiotherapy was conducted in 148 patients and 241 patients underwent surgery alone. The propensity-matched group consisted of 105 patients operated on after radiotherapy and 105 controls. The number of local recurrences was 7 (6.7%) in the preoperative radiotherapy group and 22 (21%) in the surgery alone group (*p* = 0.016). The 18-year survival analysis showed no survival benefit in the preoperative radiotherapy group (38% versus 48%, *p* = 0.107) but improved recurrence-free survival (81% versus 58%, *p* = 0.001). The preoperative short-course radiotherapy significantly decreases the risk of local recurrence in locally advanced rectal cancer and may improve recurrence-free survival without an increased risk of second primary malignancy.

## 1. Introduction

A total of one-third of colorectal cancer cases consist of rectal cancers. The last few decades have demonstrated a total reduction in local recurrence rates of rectal neoplasms. This may be due to changes in the treatment strategies and surgical techniques. The first shift of paradigm was the introduction of the total mesorectal excision (TME) technique [1]. The second was the implementation of preoperative radiotherapy. Nevertheless, surgery still remains the mainstay of rectal cancer treatment.

Clinicopathologic studies reported that most recurrences occur when the tumor spreads to the circumferential resection margin. This suggests that recurrence is closely related to the persistence of neoplastic foci within the perirectal tissues [2]. Radiotherapy aims to deliver a precise dose of ionizing radiation to a well-defined target volume. It utilizes an external beam source that delivers high-energy photons generated outside the patient. Contemporary radiotherapy uses three or more radiotherapy fields to reduce the amount of normal tissue in the target volume. The main role of radiotherapy is the treatment of microscopic disease beyond the edge of the surgical field. The theoretical advantage of neoadjuvant compared with adjuvant radiotherapy is its potential for tumor downstaging for the sake of better radical surgery outcomes and a lower risk of tumor seeding. A recently published Cochrane review showed that preoperative radiotherapy reduces overall mortality at 4–12 years of follow-up. Nevertheless, the trials with the TME technique demonstrated little effect of irradiation on patient survival [3]. Another meta-analysis revealed that short-course preoperative radiotherapy administered 4 or more weeks before the surgery was equally effective as the preoperative long-course radiotherapy in terms of overall survival and recurrence-free survival [4]. Current NCCN (National Comprehensive Cancer Network) and ESMO (European Society for Medical Oncology) guidelines propose the preoperative radiotherapy 5x5Gy as an acceptable treatment modality in locally advanced rectal carcinoma [5,6]. The majority of previously published studies have reported a decrease in local recurrence rate but not in systemic recurrence rate or survival benefit in patients who underwent neoadjuvant irradiation.

As a consequence of improved patient survival, long-term results after rectal cancer treatment are of growing interest in the medical society. A significant problem for long-term cancer survivors may be the late toxicity of radiotherapy. This creates a possible risk of developing second primary malignancy in the future and a theoretical decrease in overall survival. Few studies have analyzed the association between preoperative irradiation and secondary malignancies in rectal cancer and have found inconclusive results [7,8,9,10]. It remains unclear which treatment option (preoperative short-course radiotherapy, preoperative chemotherapy, or postoperative radiotherapy) is primarily related to an increased risk of second malignancy.

This study aimed to assess the influence of short-course preoperative radiotherapy in patients with locally advanced rectal cancer on overall survival, local recurrence rate, and second malignancy at 18-year follow-up.

## 2. Materials and Methods

The rectal cancer trial was conducted in a single tertiary center between February 1992 and June 2006. A total of 389 patients with locally advanced rectal cancer (cT2-cT4, cN0/+, cM0) were included in the study. The flow diagram of the study is shown in Figure 1.

Pretreatment T and *n* stages were determined in all patients using endorectal ultrasound, computed tomography of the pelvis and abdomen, and a chest X-ray. Preoperative radiotherapy (5 × 5 Gy delivered in 5 days) was conducted in 148 patients. The indications for preoperative radiotherapy in the study cohort included:-Primary rectal cancer cT2-cT4 cN0/+ (stage established by means of endorectal ultrasound and computed tomography)-No distant metastases (assessed by chest X-ray and abdominal computed tomography)-Tumor location below the level of S1/S2 with an inferior tumor margin 15 cm or less from the anal verge as measured during withdrawal of a rigid rectoscope.

The time interval between radiotherapy and surgery was 7–10 days (*n* = 74) or 4–5 weeks (*n* = 74). Oncological results in irradiated patients at 5-year follow-up were previously published (Krakow Rectal Cancer Trial, ClinicalTrials.gov identifier NCT01444495) [11]. The study was approved by the institutional review board (number KBET/85/B). A total of 241 patients were operated on without neoadjuvant treatment. It was not possible to include more controls in the analysis since preoperative radiotherapy has become the standard treatment of locally advanced rectal cancer after the Swedish Rectal Cancer Trial. The control group used in propensity score matching was a unique group of patients with locally advanced cancer, who were operated on without neoadjuvant treatment and were followed-up on. Patients in both groups were operated on according to the same standard, i.e., total mesorectal excision. Radical resection (R0) was performed in 133 patients after irradiation and in 220 patients without radiotherapy. The clinicopathological features of the analyzed patients are summarized in Table 1.

The groups were heterogeneous in terms of pretreatment nodal status. In addition, the imbalance of potential confounders between the radiotherapy and surgical resection groups was addressed by matching the treatment groups using propensity scores. The only curatively treated patients (R0) were matched. The characteristics of matched groups are summarized in Table 2.

Anterior resection was performed only when a 2 cm margin below the distal tumor margin was achievable. In other cases, an abdominoperineal resection approach was chosen. A high ligation of the inferior mesenteric artery proximal to the origin of the left colic artery was performed in all patients. The staging was established according to the American Joint Committee on Cancer (AJCC, seventh edition) and marked as (y)pTNM.

Chemotherapy was initiated 4 weeks after the surgery. Patients with cancer stage III, IV, or II with clinicopathologic features associated with worse prognosis (T4 tumor, high-grade histology, signet ring or mucinous tumor, bowel obstruction or perforation) were qualified for either the standard adjuvant chemotherapy with 5-fluorouracil and leucovorin (Mayo Clinic regimen) or chemotherapy with irinotecan, 5-fluorouracil and leucovorin (CLF regimen). Chemotherapy was not administered to patients with complete pathological response, cancer stage I, or cancer stage II without features associated with a worse prognosis, Eastern Cooperative Oncology Group (ECOG) performance status >2, and those who refused to undergo systemic therapy.

Patients were evaluated every 3 months during the first 2 years, then every 6 months during the next 3 years, and then yearly at the outpatient department. Rectoscopy, endorectal ultrasound, and an abdominal ultrasound were performed every 6 months, a chest X-ray once a year, and a colonoscopy after 1, 3, and 5 years. After 5 years, the patients with symptoms suggesting recurrence were thoroughly evaluated at the outpatient department. Recurrences were confirmed by computed tomography. During each follow-up visit, the patient’s history dealing with urogenital symptoms was taken and a physical examination was performed. If suspicion of any disease of the urogenital tract was raised, an appropriate consultation was performed (urological or gynecological) to exclude potential secondary malignancy in the irradiated area. Dates of death were obtained from the national census registry office. Survival rates were calculated based on the overall survival principle; deaths due to any cause were accepted as complete observations, while lack of follow-up was considered as a censored observation. The median follow-up of surviving patients was 18.0 years. Overall survival (OS) was calculated from the day of surgery. Recurrence-free survival (RFS) was defined as the time from the day of surgery to the first event of either death or recurrent disease. Only patients with curative resection (R0) were included in the recurrence and survival analyses.

The primary endpoint was the local recurrence rate. Secondary endpoints were overall survival, recurrence-free survival, and rate of second malignancy.

This study size was determined by the number of patients operated on with locally advanced rectal cancer. We compared the baselines and matched the characteristics using standard tests for categorical variables and variables without normal distribution (χ² and U Mann–Whitney tests).

To address the imbalance of potential confounders between the preoperative radiotherapy and surgical resection groups, we matched treatment groups using propensity scores. The propensity score was estimated based on the predicted probability of a patient being in the preoperative radiotherapy group in a logistic regression model. The propensity score model included the cT stage, cN stage, the distance from the anal verge, and age. Matched pairs were formed between patients treated by preoperative radiotherapy and those who had surgical resection alone using a one-to-one nearest neighbor caliper of width 0.1 (maximum allowable difference in propensity scores). Only patients matched with propensity scores were included in the time-to-event analyses.

The predicted local recurrence rate for the preoperative radiotherapy and surgery was estimated as 5%, whereas the overall recurrence rate for surgery alone was estimated as being 20%. To achieve a test power of 80%, the sample size needed for each arm was established as 70 patients (two-sided test, α = 0.05). The calculation included an anticipated drop-out rate of 1% and ARCSINUS approximation. A GRANMO sample size calculation software version 7.12 was used (https://www.imim.es/ofertadeserveis/en_granmo.html, accessed on 1 March 2021).

The Kaplan–Meier curves were constructed for all time-to-event endpoints, taking time zero as the date of surgery, and determined survival estimates with 95% confidence intervals.

To assess the differences between groups in recurrence-free survival and overall survival, the Cox models were used. All statistics were calculated with IBM SPSS Statistics version 27 for Mac (IBM Corp., Armonk, NY, USA).

## 3. Results

The matched group consisted of 105 patients operated on after short-course preoperative radiotherapy and 105 patients who only underwent surgical treatment.

### 3.1. Recurrences

The number of local recurrences was 7 (6.7%) in the preoperative radiotherapy group and 22 (21%) in the surgery-alone group (*p* = 0.016). The majority of local recurrences occurred within five years after the surgery (5 and 19, respectively). Between 5–10 years, only two local recurrences were diagnosed in irradiated patients and there were three local recurrences in the surgery-alone group. No local recurrences were observed >10 years from surgery. Patients in the preoperative radiotherapy group had a 72% lower risk of local recurrence (Hazard ratio HR = 0.278 (95% CI 0.119–0.652; *p* = 0.003). When both systemic and local recurrences were analyzed, patients after preoperative radiotherapy had a 60% lower risk of recurrence (HR 0.395, 95% CI 0.230–0.677; *p* = 0.001). The data on recurrences are summarized in Table 3.

The univariate analysis has identified a higher ypN stage, >12 lymph nodes retrieval, localization < 5 cm from the anal verge, and surgery without preoperative radiotherapy as factors that increase the risk for local recurrence. In multivariate analysis, two independent risk factors that decrease the risk of local recurrence were: preoperative radiotherapy (OR 0.212, 95% CI 0.088–0.512) and tumor location 5–10 cm from the anal verge (OR 0.299, 95% CI 0.143–0.625) or >10 cm from the anal verge (OR 0.085, 95% CI 0.011–0.674). The data on risk factors influencing local recurrence are summarized in Table 4.

### 3.2. Overall Survival

Overall survival in all patients is presented in Figure 2A. Overall survival for (y)pTNM stages I, II, and III are presented in Figure 2B–D, respectively. The 18-year survival analysis showed no survival benefit in the preoperative radiotherapy group (38% versus 48%, *p* = 0.107, log-rank test). Moreover, no survival benefit was observed when comparing OS separately for stages I, II, and III. The patients in the radiotherapy group had a similar risk of death as those without radiotherapy (HR = 0.740, 95% CI 0.513–1.068; *p* = 0.108). No second malignancies were diagnosed within pelvic organs and tissues in either of the groups during the follow-up.

### 3.3. Recurrence-Free Survival

The recurrence-free survival for all patients is presented in Figure 3A. The recurrence-free survival for (y)pTNM stages I, II, and III are presented in Figure 3B–D, respectively. The 18-year survival analysis showed the recurrence-free survival benefit in the preoperative radiotherapy group (81% versus 58%, *p* = 0.001, log-rank test). The recurrence-free survival benefit was observed for stage I patients (92% versus 68%, *p* = 0.014, log-rank test) but not for stage II and stage III patients (*p* = 0.184 and *p* = 0.077, respectively).

## 4. Discussion

The value of short-course preoperative radiotherapy has already been widely investigated in the past. However, there is still little research on the oncological results at follow-up longer than 10 years. The goal of this study was to elaborate on the effects of short-course preoperative radiotherapy after 18 years of patient follow-up. Our study included 389 patients operated on in a single tertiary center and analyzed them after propensity score matching. The surgeries were conducted by the total mesorectal excision technique but before the quality-controlled TME technique was commonly introduced. Consequently, we cannot provide exact data on the quality of mesorectal excision according to the classification of Quirke et al. [12] The quality of resection in our patients may be confirmed by the lymph node yield with the median value of 16, even in those operated after irradiation.

Our study showed that preoperative radiotherapy 5 × 5 Gy decreased the local recurrence rate from 21% to 6.7%. The hazard ratio of local recurrence was significantly reduced after neoadjuvant therapy (*p* = 0.003). The preoperative irradiation and tumor localization > 5 cm from the anal verge have contributed to the decrease in the rate of local recurrence, as shown in multivariate analysis.

Similar results were reported in Swedish Rectal Cancer Trial, in which the local recurrence rate decreased from 26% to 9% in patients operated after preoperative irradiation [13]. The study has investigated patients operated before quality-controlled TME surgery and results are available for the median follow-up of 13 years. In the Dutch TME Trial, neoadjuvant radiotherapy 5 × 5 Gy reduced 10-year local recurrence by more than 50% relative to the surgery alone [14]. The 10-year incidence of local recurrence in the latter study was 3% in the radiotherapy-surgery group and 9% in the surgery-alone group (*p* < 0.0001). The Stockholm III trial analyzed the influence of 25 Gy radiotherapy with standard (within 1 week) or delayed (4–8 weeks) surgery and long-course radiotherapy (25 × 2 Gy without chemotherapy) on local recurrence rate. The study showed that both time intervals for administration of short-course preoperative irradiation were deemed as non-inferior results compared to the long-course treatment with a delay of 4–8 weeks (15). Furthermore, Trans-Tasman Radiation Oncology Group trial 01.04 showed that for cT3 rectal cancer 3-year local recurrence rates between 5 × 5 Gy and that preoperative chemoradiotherapy (50.4 Gy) were not significantly different (*p* = 0.24) [15].

On the other hand, some research supports the notion that radiotherapy may not be necessary for selected patient groups. The data from the Norwegian trial published in 2009 showed a 5-year local recurrence of 7% in patients without neoadjuvant treatment [16].

The majority of the above-mentioned studies did not report a detailed analysis of risk factors of local recurrence. Our results suggest that apart from radiotherapy, the distance from the anal verge seems to be crucial for the local recurrence rate. The Swedish Rectal Cancer Trial revealed that local recurrence rates were significantly lower after radiotherapy when the tumour was localized in the low (*p* = 0.003) and middle (*p* < 0.001) part of the rectum. As a comparison, our study revealed a lower rate of local recurrence in patients with tumors in the middle and upper part of the rectum.

Further research is needed to unequivocally establish the value of preoperative treatment in locally advanced rectal cancer.

The preoperative short-course radiotherapy did not affect overall survival or cancer-specific survival in the studies comparing TME surgery alone with neoadjuvant irradiation followed by TME [14,17]. The Swedish Rectal Cancer Trial showed that preoperative irradiation (5 × 5 Gy) followed by surgery within 1 week improved overall survival after a 5-year and 13-year follow-up, which may be attributable to a significant decrease in local recurrence [13]. In our study, no survival benefit was observed at 18-year follow-up in propensity-matched groups. Patients were operated on according to the TME technique, but the quality of the specimen was not assessed as proposed by Quirke et al. in 2002 [12]. Therefore, we do not have exact data on the percentage of mesorectal plane resections and the quality of performed procedures may be evaluated only with the median number of retrieved lymph nodes. Currently, the optimal quality-controlled surgery can be associated with a low local recurrence rate (<10%) even in patients without preoperative radiotherapy [18]. The question was raised as to whether radiotherapy is needed at all in locally advanced rectal cancer. Our study supports the use of neoadjuvant irradiation because the analyzed patients, who underwent radical resection (R0), benefitted from a lower local recurrence rate, and had improved recurrence-free survival. Moreover, the difference in disease-free survival was statistically significant in stage I patients. In the Dutch TME trial, preoperative radiotherapy improved cancer-specific survival in patients operated on with a negative circumferential resection margin. Besides, it showed that for TNM III patients, only 10 patients have to be treated with radiotherapy to save one life. No survival benefit was reported for TNM stage I and II patients [14].

In the present study, a recurrence-free survival benefit was observed for all irradiated patients, which was attributable to ypTNM I patients. This result suggests that radiotherapy can play an important role mainly in those individuals in whom a response to irradiation was observed. Although disease-free survival is evaluated in oncological trials with a shorter follow-up, the difference in DFS does not necessarily correspond to different overall survival, especially when long-term results are analyzed. Similarly, our study revealed no overall survival benefit for TNM stage I–III patients.

No other neoplasms were diagnosed in the analyzed patient group. Despite the potential risk of late side effects of radiotherapy and the risk of second primary malignancy, irradiation did not decrease survival at 18-year follow-up. The previous studies that addressed the issue had divergent results. Some revealed no increased risk of second primary malignancy in irradiated patients [9,19,20]. Others reported the increased risk of second cancer (relative risk 1.15) seen in organs within or adjacent to the irradiated volume (relative risk 2.04) but not outside the irradiated volume [7,8]. Our study confirms that short-course preoperative radiotherapy is safe in long-term cancer survivors.

We used propensity score matching to estimate the effect of radiotherapy on survival outcomes. A potential limitation of this method is the requirement for the pool of controls. There were only 241 patients operated on according to the TME technique without previous radiotherapy and they were matched to irradiated patients. The advantage of matching was a better transparency than the inverse probability of treatment weighting, which required the creation of a synthetic weighted sample. However, neither of the approaches was superior to the other. We conducted caliper matching (width 0.1) that enabled the elimination of a greater degree of systematic differences between the analyzed groups. As a consequence, fewer patients were matched. Despite the limitations mentioned above, the propensity score matching created two samples with comparable baseline characteristics and enabled an analysis of long-term oncological outcomes.

## 5. Conclusions

The preoperative short-course radiotherapy significantly decreases the risk of local recurrence in locally advanced rectal cancer and may improve recurrence-free survival in the follow-up period of 18 years.

## Figures and Tables

**Figure 1 biomedicines-09-00725-f001:**
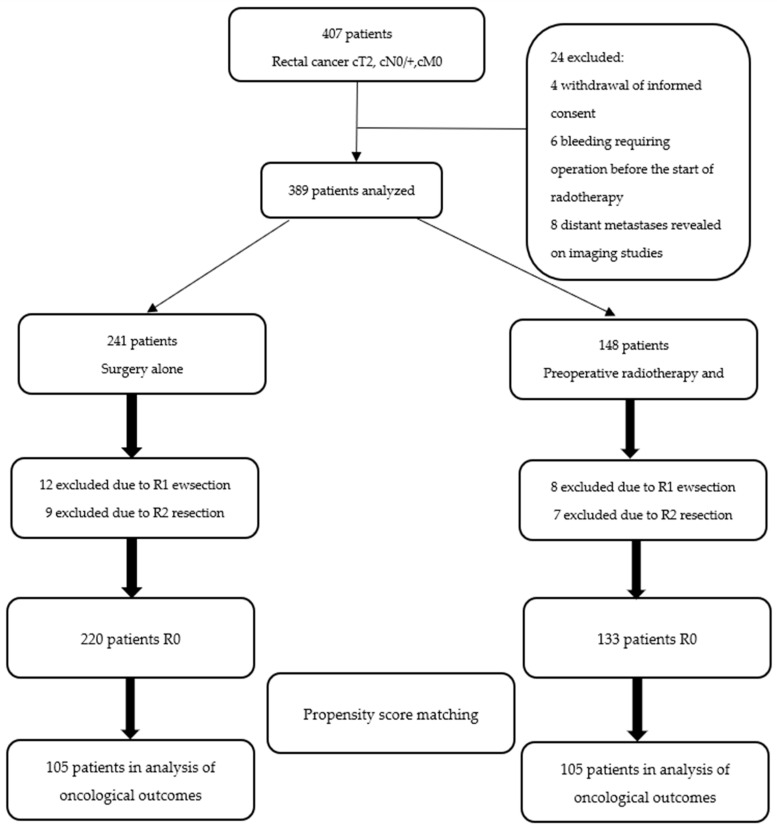
Flow-diagram of the study.

**Figure 2 biomedicines-09-00725-f002:**
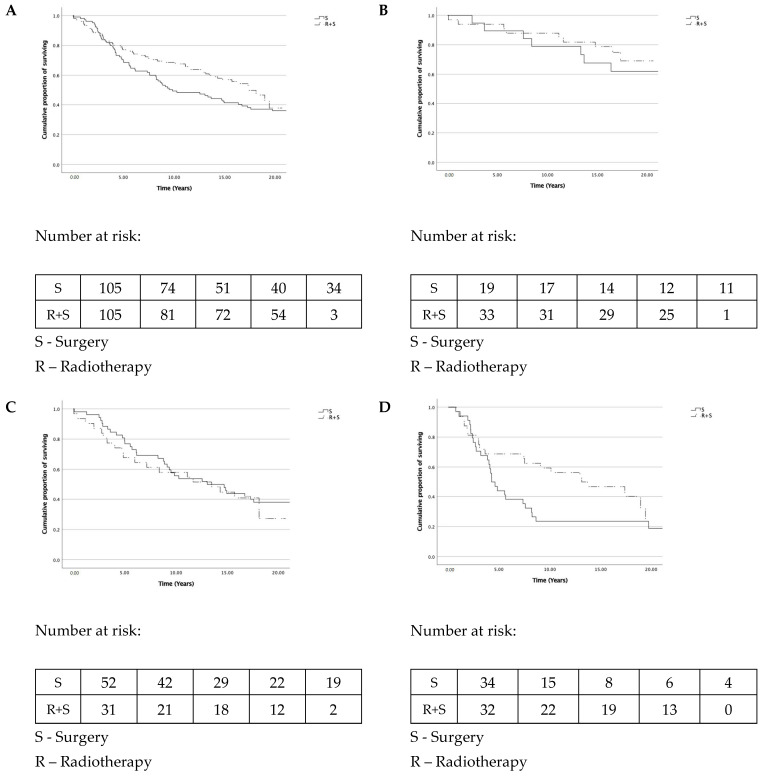
Overall survival. (**A**) For all patients, log-rank *p* = 0.107. (**B**) For stage (y)pI patients, log-rank *p* = 0.513. (**C**) For stage (y)pII patients, log-rank *p* = 0.616. (**D**) For stage (y)pIII patients, log-rank *p* = 0.094.

**Figure 3 biomedicines-09-00725-f003:**
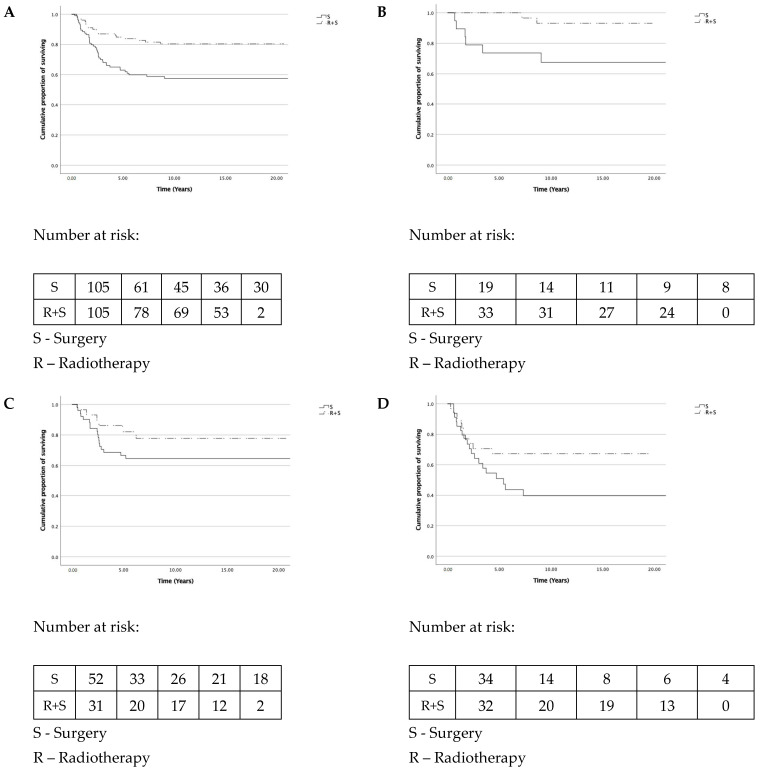
Recurrence-free survival. (**A**) For all patients, log-rank *p* = 0.001. (**B**) For stage (y)pI patients, log-rank *p* = 0.014. (**C**) For stage (y)pII patients, log-rank *p* = 0.184. (**D**) For stage (y)pIII patients, log-rank *p* = 0.077.

**Table 1 biomedicines-09-00725-t001:** Patient clinical and demographic characteristics, by treatment group before propensity-score matching (R0 patients).

	Surgery Alone	Preoperative Radiotherapy and Surgery	*p* Value
	*n* = 220	*n* = 133	
Sex			0.644 *
Men	123 (56%)	71 (53%)	
Women	97 (44%)	62 (47%)	
Age (years, median, minimum-maximum)	65 (36–86)	62 (26–92)	0.163 †
Pretreatment tumor stage (T)			0.815 *
cT2	46 (21%)	26 (20%)	
cT3	161 (73%)	97 (73%)	
cT4	13 (6%)	10 (8%)	
Pretreatment nodal status (*n*)			0.0005 *
cN0	136 (62%)	106 (80%)	
cN+	84 (38%)	27 (20%)	
Height from anal verge			0.055 *
<5 cm	46 (21%)	38 (29%)	
5–10 cm	128 (58%)	79 (59%)	
>10 cm	46 (21%)	16 (12%)	
Chemotherapy received			0.836 *
Yes	143 (65%)	85 (64%)	
No	77 (35%)	48 (36%)	
Lymph node yield (median, minimum-maximum)	16 (0–68)	16 (0–79)	0.406 †

* χ² test or Fisher’s exact test. † U Mann–Whitney test.

**Table 2 biomedicines-09-00725-t002:** Patient clinical and demographic characteristics, by treatment group after propensity-score matching.

	Surgery Alone	Preoperative Radiotherapy and Surgery *n* = 105	*p* Value
	*n* = 105		
Sex			1.000 *
Men	61 (58%)	61 (58%)	
Women	44 (42%)	44 (42%)	
Age (years, median, minimum-maximum)	64 (36–80)	61 (26–78)	0.222 †
Pretreatment tumor stage (T)			0.549 *
cT2	25 (24%)	22 (21%)	
cT3	74 (70%)	73 (70%)	
cT4	6 (6%)	10 (9%)	
Pretreatment nodal status (*n*)			0.287 *
cN0	71 (68%)	78 (74%)	
cN+	34 (32%)	27 (26%)	
Pretreatment stage (cUICC)			0.537 *
I	19 (18%)	19 (18%)	
II	52 (50%)	59 (56%)	
III	34 (32%)	27 (26%)	
Height from anal verge			0.897 *
<5 cm	25 (24%)	23 (22%)	
5–10 cm	66 (63%)	66 (63%)	
>10 cm	14 (13%)	16 (15%)	
Chemotherapy received			0.558 *
Yes	37 (35%)	33 (31%)	
No	68 (65%)	72 (69%)	
Lymph node yield (median, minimum-maximum)	17 (0–68)	16 (0–79)	0.481 †

* χ² test or Fisher’s exact test. † U Mann–Whitney test.

**Table 3 biomedicines-09-00725-t003:** Oncological outcomes in analyzed groups (R0), comparison at the end of follow-up.

	Surgery	Radiotherapy + Surgery	*p* Value
	*n* = 105	*n* = 105	
Local recurrence			
HR (95% CI)	1.00 (ref)	0.278 (0.119–0.652)	0.003
Overall survival			
HR (95% CI)	1.00 (ref)	0.740 (0.513–1.068)	0.108
Recurrence-free survival			
HR (95% CI)	1.00 (ref)	0.395 (0.230–0.677)	0.001

HR-hazard ratio, Cox regression, log-rank test. CI-confidence interval.

**Table 4 biomedicines-09-00725-t004:** Univariate and multivariate analysis of risk factors of local recurrence.

Variable	Description	Univariate Analysis		Multivariate Analysis	
		OR (95% CI)	*p* *	OR (95% CI)	*p* *
Sex	Male	1	0.455		
	Female	1.350 (0.615–2.963)			
Age	≤65 years	1	0.707		
	>65 years	0.854 (0.375–1.944)			
ypT stage	0–2	1	0.200		
	3–4	1.804 (0.731–4.449)			
ypN stage	0	1	0.001	1	0.726
	1	8.225 (3.828–17.672)		1.450 (0.417–5.044)	
	2	3.804 (1.863–7.769)		1.436 (0.450–4.578)	
Number of harvested LNs	<12	1	0.001	1	0.604
	≥12	0.242 (0.129–0.453)		0.766 (0.280–2.098)	
Radiotherapy	No	1	0.001	1	0.001
	Yes	0.071 (0.033–0.154)		0.212 (0.088–0.512)	
Tumour location	<5 cm	1	0.001	1	0.001
	5–10 cm	0.158 (0.096–0.260)		0.299 (0.143–0.625)	
	>10 cm	0.034 (0.005–0.253)		0.085 (0.011–0.674)	
Chemotherapy	No	1	0.161		
	Yes	0.566 (0.255–1.254)			

* logistic regression. OR-odds ratio, CI-confidence interval.

## Data Availability

The data presented in this study are available on request from the corresponding author. The data will be publicly available after anonymization.

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
