# Peer review of "Preoperative Short-Course Radiotherapy and Surgery versus Surgery Alone for Patients with Rectal Cancer: A Propensity Score-Matched Analysis at 18-Year Follow-Up"

_biomedicines, 2021, doi:10.3390/biomedicines9070725_

Round 1

Reviewer 1 Report

The authors of the manuscript »Preoperative short-course radiotherapy and surgery versus surgery alone for patients with rectal cancer: a propensity score-matched analysis at 18-year follow-up« provide a long term follow-up of the study conducted in 1992 to 2006. The subject of the manuscript is interesting since the issue of neoadjuvant treatment of locally advanced rectal cancer is still evolving, which is exactly the point the authors deal with very nicely in the introduction section. In the methods section they describe the study design and present the patients demographics. They explain nicely how they used the propensity score matching to deal with stage discrepancy between irradiated and non-irradiated patients. They however do not provide information on indications used for preoperative radiotherapy in the study cohort. This would be of particular interest in the light of rather high percentage of stage I irradiated patients. The follow-up after surgery was appropriate for the time period studied, yet the authors do not provide data on how the potential secondary malignancies were actively sought for. In the results section they nicely provide data on local recurrences, overall and recurrence-free survival. In the discussion section they compare their results with relevant other studies. Interestingly they point out the benefit of preoperative radiotherapy in terms of better recurrence-free survival for the group of stage I patients who should have not received the treatment by any recommendation in the first place. Furthermore of interest is the fact that the preoperative radiation benefit in terms of better recurrence-free survival for all stages doesn’t seem to translate into better overall survival after 18 years, despite the fact, that all recurrences occurred within 10 years. The conclusions they made are however fair. The references are relevant.

Author Response

Thank you for all comments to our manuscript. Our responses are written below after the Reviewer's comments. We modified the manuscript accordingly - all added sentenced are marked in yellow in the revised version of the manuscript. 

  • They however do not provide information on indications used for preoperative radiotherapy in the study cohort. This would be of particular interest in the light of rather high percentage of stage I irradiated patients.

The indications for preoperative radiotherapy in the study cohort included:

  • Primary rectal cancer cT2-cT4 cN0/+ (stage established by means of endorectal ultrasound)
  • No distant metastases (assessed by chest X-ray and abdominal CT)
  • Tumour location below the level of S1/S2 with inferior tumour margin 15 cm or less from the anal verge as measured during withdrawal of a rigid rectoscope

We added abovementioned criteria to the manuscript.

  • The follow-up after surgery was appropriate for the time period studied, yet the authors do not provide data on how the potential secondary malignancies were actively sought for.

During each follow-up visit patient’s history dealing with urogenital symptoms was taken and physical examination was performed. If suspicion of any disease of urogenital tract was raised, appropriate consultation was performed (urological or gynecological) to exclude potential secondary malignancy in the irradiated area.  We added this piece of information to the manuscript.

  • Interestingly they point out the benefit of preoperative radiotherapy in terms of better recurrence-free survival for the group of stage I patients who should have not received the treatment by any recommendation in the first place.

Recurrence-free survival and overall survival curves were drawn for stages established in histopathological examination of the specimen (ypUICC). Therefore, there were 19 irradiated patients with clinical stage cI but 33 patients with stage ypI. The group ypI included patients with downstaging after radiotherapy as well. We emphasized the issue in the discussion stating that in the present study, recurrence-free survival benefit was observed for all irradiated patients, which was attributable to ypTNM I patients. To clarify this issue, we added “yp” description in the results section of the manuscript.

  • Furthermore of interest is the fact that the preoperative radiation benefit in terms of better recurrence-free survival for all stages doesn’t seem to translate into better overall survival after 18 years, despite the fact, that all recurrences occurred within 10 years.

Thank you for raising this issue. We commented on it in the discussion of the original contribution: “Although disease-free survival is evaluated in oncological trials with shorter follow-up, the difference in DFS does not necessarily correspond with different overall survival especially when long-term results are analyzed. Similarly, our study revealed no overall survival benefit for TNM stage I - III patients.”. It may be hypothesized that better recurrence-free survival does not translate into better overall survival because in long follow-up other causes of death than disease relapse are observed. On the other hand, some patients with recurrence underwent radical resection and their survival was not influenced by disease relapse. 

Reviewer 2 Report

Pach et al. in their manuscript entitled "Preoperative short-course radiotherapy and surgery versus surgery alone for patients with rectal cancer: a propensity score-matched analysis at 18-year follow-up" elegantly present an in-depth analytical comparison between radiotherapy-surgery combination and surgery alone management strategy for rectal cancer patients. 

Author Response

Thank you very much for your positive opinion. We are very pleased that our efforts to prepare this manuscript with long follow-up results, have been appreciated.